# The Impact of Cannabidiol (CBD) on Lipid Absorption and Lymphatic Transport in Rats

**DOI:** 10.3390/nu17061034

**Published:** 2025-03-15

**Authors:** Qi Zhu, Qing Yang, Ling Shen, Meifeng Xu, Min Liu

**Affiliations:** Department of Pathology and Laboratory Medicine, University of Cincinnati College of Medicine, Cincinnati, OH 45237, USA; zhuqu@ucmail.uc.edu (Q.Z.); yangqa@ucmail.uc.edu (Q.Y.); shenln@ucmail.uc.edu (L.S.); xume@ucmail.uc.edu (M.X.)

**Keywords:** CBD, lipid absorption and transport, lymph fistula model, apolipoprotein

## Abstract

**Background:** Cannabidiol (CBD) exerts diverse metabolic effects, yet its influence on intestinal lipid metabolism remains unclear. **Methods:** In this study, we investigated whether short-term (one-week) CBD treatment affects lipid absorption and transport through the lymphatic system using a validated lymph fistula model. **Results:** CBD treatment significantly enhanced the transport of radiolabeled triglycerides through the lymphatic system. This effect appeared specific, as CBD did not substantially alter cholesterol output in the lymph. Chemical assays indicated that CBD treatment did not significantly alter total triglycerides, cholesterol, phospholipids, or non-esterified fatty acid levels in the lymph. However, it significantly enhanced the lymphatic output of apolipoprotein A4 (ApoA4) and apolipoprotein A1 (ApoA1). Additionally, gene expression analysis revealed a downregulation of vascular endothelial growth factor receptor 1 (Flt1) in the small intestine, leading to increased lymphatic lacteal permeability and altered lipid transport dynamics. **Conclusions:** These findings indicate that short-term CBD treatment modulates lymphatic lipid composition and apolipoprotein secretion by regulating lymphatic lacteal function, thereby influencing lipid transport and metabolism. This study provides novel insights into CBD’s role in facilitating TG-rich lipoprotein transport via the lymphatic system, highlighting its potential therapeutic applications in lipid-related disorders.

## 1. Introduction

Cannabidiol (CBD), a non-psychoactive phytocannabinoid, is widely recognized for its anti-inflammation, analgesic, and anti-depressant properties [1,2]. Recent research has also highlighted its therapeutic potential in metabolic diseases, including diabetes [3]. Despite growing interest, the anti-obesity effect of CBD remains incompletely understood. Studies have shown that CBD reduces body weight gain in rats via cannabinoid receptor 2 (CB2) receptors [4] and inhibits lipogenesis and promotes browning in 3T3-L1 adipocytes [5]. Additionally, CBD modulates inflammation and alters gut microbiota composition, which may influence small intestinal permeability and the overall intestinal microenvironment, thereby impacting nutrient absorption [6]. Moreover, CBD improves lipid metabolism and boosts mitochondrial activity, leading to reduced hepatic fat accumulation in zebrafish and obese mouse models [7]. Given its role in lipid metabolism across central and peripheral tissues, CBD may also impact lipid absorption in the small intestine. Understanding this influence is essential, as lipid absorption directly affects energy balance, nutritional status, and the pharmacokinetics of lipid-soluble drugs and bioactive compounds, including CBD itself.

Previous findings showed that dietary lipids enhance CBD absorption following oral administration [8,9]. However, the specific effects of CBD on lipid absorption and transport remain unexplored. Furthermore, existing research has primarily focused on lipoprotein profiles in plasma, neglecting direct lymphatic measurements, likely due to the technical challenges of lymph sampling in human trials. Consequently, it remains unclear how CBD treatment influences lymphatic lipid profiles. Investigating lipid absorption through the lymphatic system is critical, as it provides a direct assessment of lipid uptake and secretion from the small intestine compared to plasma-based analyses.

Lacteals, specialized lymphatic capillaries in the small intestine, exhibit structural changes that influence lipid metabolism. Their endothelial junctions change from button-like to zipper-like formations, or vice versa. Button junctions dominate active lipid absorption, allowing chylomicrons to efficiently enter the lymphatic system. In contrast, the zipper junction lowers lacteal permeability and restricts lipid entry. Certain genes, such as vascular endothelial growth factor receptor 1 (Flt1), regulate the switch between these structures to optimize fat absorption [10]. We hypothesize that the CBD treatment increases lipid output in lymph, potentially due to the modification of lacteal function. To test this hypothesis, we utilized a conscious rat model with mesenteric lymph duct cannulation. This approach allowed for the direct measurement of the lymph flow rate, lipid transport through the lymphatic system, apolipoprotein secretion, and the expression of genes linked to lymphatic lacteal function.

## 2. Methods

### 2.1. Animals

Adult male Sprague Dawley rats (300–350 g) were obtained from Envigo (Indianapolis, IN, USA) and housed in the animal facility for a two-week acclimation period before experimentation. All animal procedures were conducted in accordance with a protocol approved by the University of Cincinnati Institutional Animal Care and Use Committee and complied with the National Institutes of Health Guide for the Care and Use of Laboratory Animals.

### 2.2. CBD Treatment Study

CBD Isolate (>99% purity) was obtained from Impact Naturals, Inc. (San Diego, CA, USA) and dissolved in sesame oil, an appropriate solvent for CBD [8]. Rats received subcutaneous injections twice daily for one week with either vehicle (sesame oil), low-dose CBD (10 mg/kg), or high-dose CBD (30 mg/kg). These doses are within the range commonly used in preclinical studies to investigate CBD’s effects on lipid metabolism, inflammation, and other physiological processes [11,12]. The body weight of each group of rats was recorded every two days.

### 2.3. Lipid Infusate Preparation

Chloroform was used to dissolve non-radioactive triolein (360 mg), [^3^H]-labeled triolein (10 μCi), cholesterol (30 mg), [^14^C]-labeled cholesterol (1 μCi), and egg phosphatidylcholine (67.4 mg). The solvent was then evaporated under a steady stream of nitrogen, leaving a chloroform-free lipid mixture. This mixture was emulsified in 30 mL of phosphate-buffered saline (PBS; 6.74 mM Na_2_HPO_4_, 16.5 mM NaH_2_PO_4_, 115 mM NaCl, and 5 mM KCl, pH 6.4) containing 19 mM sodium taurocholate (NaTC). The mixture was sonicated with alternating 1-s on/off pulses until a homogeneous solution was obtained. The emulsion’s uniformity was confirmed by measuring the radioactivity at the top, middle, and bottom, ensuring variation did not exceed 1% [13].

### 2.4. Lymph Collection

Rats underwent surgery for the implantation of lymphatic and gastric cannulas, as was conducted previously [9]. Briefly, after an overnight fast, rats were anesthetized with isoflurane, and a midline vertical incision was made. The superior mesenteric lymphatic duct was cannulated with polyvinylchloride (PVC) tubing. A silicone feeding tube was inserted into the duodenum through a stomach incision. Following surgery, the animals were continuously infused overnight through a feeding tube with a saline solution containing 5% glucose at a rate of 3 mL/h to compensate for the fluid and electrolyte loss caused by lymphatic drainage. They were allowed to recover overnight in Bollman restraining cages, housed in a temperature-regulated chamber (~28 °C) to maintain warmth and comfort. Although the animals were restrained, the animals retained significant mobility, allowing them to move backward, forward, and sideways.

After overnight recovery, fasting lymph was collected and kept on ice for 1 h prior to lipid infusion. Each rat then received a subcutaneous injection of either a matched dose of CBD or the vehicle. One hour later, the lipid emulsion was continuously administered into the duodenum at a constant rate of 3 mL/h for a duration of 6 h. Lymph samples were collected hourly for 6 h into pre-cooled conical centrifuge tubes. A 100 μL aliquot from each lymph sample was mixed with Opti-Fluor (Packard Bioscience, Meriden, CT, USA) and analyzed using a liquid scintillation spectrometer (LS 6500 multi-purpose scintillation counter, Beckman Coulter). The output of labeled lipids into the lymph was calculated each hour based on the corresponding lymph volume or flow rate. Using the infused amount per hour, we calculated the lymphatic output as a percentage of the total infused dose.

After the 6 h period, the rats were euthanized, and the stomach, small intestine, and colon were collected. The luminal contents of these tissues were obtained by washing them three times with 5 mL of a 10 mM NaTC solution. The small intestine was divided into four equal-length segments, designated as M1–M4: M1 (duodenum), M2 and M3 (two equal segments of the jejunum), and M4 (ileum). Lipids were extracted from each segment following the method described by Folch et al. [14]. An aliquot of each sample was mixed with Opti-Fluor and analyzed using a liquid scintillation spectrometer. The total count for each sample was calculated based on its total volume and the measured aliquot count from the scintillation counter.

### 2.5. Calculations of Absorptive and Transport Index

The absorptive and lymph transport indexes were assessed using previously established methods [15]. The absorptive index, which represents the percentage of infused lipid absorbed by enterocytes, was calculated using the following formula: Absorptive Index = 100% − % total dose recovered in the gastrointestinal lumen. The lymphatic transport index measures the proportion of absorbed lipid secreted into the lymph, accounting for intestinal lipid uptake. It was calculated as follows: Lymphatic Transport Index = % infused dose recovered in lymph/Absorptive Index × 100. This index serves as a valuable measure of the small intestine’s efficiency in transporting lipids into lymph as chylomicrons (CMs).

### 2.6. Quantification of TG, Cholesterol, Phospholipids, and Non-Esterified Fatty Acid

The TG concentration was measured using Total TG assay kits (Randox Laboratories, Kearneysville, WV, USA), and CHOL was determined by chemical assay with Infinity CHOL Liquid Stable Reagent (Fisher Diagnostics, Thermo Scientific, Middletown, VA, USA). Phospholipids (PLs) were assessed using a chemical assay with Phospholipids C reagent (Wako Diagnostics, Mountain View, CA, USA), and non-esterified fatty acid (NEFA) was measured using the HR series NEFA-HR (2) Reagent (FUJIFILM Medical Systems USA, Miami, FL, USA). All assays were conducted following the manufacturer’s instructions. Lipid output was calculated by multiplying the lymph flow rate by the corresponding lipid (TG, CHOL) measurements obtained from the scintillation counter. The trapezoid method was used to calculate the area under the curve (AUC) following treatment.

### 2.7. Measurements of Apolipoproteins

Apolipoproteins A1, A4, and B48 were quantitated by Western blotting. Two microliters of lymph samples were added to 8 µL sample buffer and loaded onto 4–15% polyacrylamide gradient gels and subsequently transferred to polyvinylidene difluoride (PVDF) membranes. After transfer, the membranes were blocked for 1 h with 5% nonfat milk in Tris-buffered saline (TBS) containing 0.1% Tween 20. They were then incubated overnight with antibodies against ApoA1 (1:5000), ApoA4 (1:5000), and ApoB48 (1:8000). After washing with TBS, the membranes were incubated for 30 min with a peroxidase-conjugated anti-goat secondary antibody diluted 1:10,000. Protein bands were visualized using Immobilon Western Chemiluminescent HRP substrate (EMD Millipore, Billerica, MA, USA). Images of the reacted membranes were captured, and band density was analyzed using the ChemiDoc Imaging System (Bio-Rad Laboratories, Inc., Hercules, CA, USA). Fold changes in the lymphatic apolipoprotein output, including ApoA1, ApoA4, and ApoB48, were determined by normalizing the relative lymph apolipoprotein levels at each hourly time point to their respective baseline (0 h, fasting) levels.

### 2.8. Quantitative Real-Time PCR (qPCR) for the Expression of Target Genes

Total RNA was isolated from the jejunum using the Qiagen RNA extraction kit (Qiagen, Germantown, MD, USA). Subsequently, 200 ng of total RNA from each sample was reverse transcribed into cDNA using the Transcriptor First Strand cDNA Synthesis Kit (ThermoFisher Scientific, Middletown, VA, USA). The mRNA levels of target genes, including vascular endothelial growth factor receptor 1 (*Flt1*) and Neuropilin1 (*Nrp1*), were measured by qPCR using TaqMan Fast Advanced Master Mix and TaqMan Gene Expression Assays on a *StepOne^TM^ Plus* device (ThermoFisher Scientific). β-actin mRNA levels were used as internal controls to normalize the gene expression data.

### 2.9. Statistics

Data are expressed as the mean ± SEM (standard error of the mean). One-way ANOVA was used for single-factor comparisons. A two-way repeated measures ANOVA, followed by Tukey’s post hoc test, was performed to analyze comparisons involving two independent variables. All statistical analyses were conducted using GraphPad Prism V8, with significance defined as *p* < 0.05

## 3. Results

### 3.1. Body Weight and Lymph Flow Change

Despite a slight downward trend in body weight in the high-dose CBD group, all three groups maintained consistent body weight throughout the treatment period (Figure 1a). The fasting lymph flow ranged from 2.2 to 2.7 mL/h across groups (Figure 1b). In the vehicle group, the lymph flow decreased at hour two and remained stable at 2.2 mL/h. In the low-dose CBD group, the lymph flow initially declined to 1.9–2.0 mL/h within 1–2 h post-infusion before stabilizing at 2.2 mL/h. The high-dose CBD group showed an initial increase in the first hour, a decline in the second hour, and stabilization at 2.0 mL/h from hour three onward. No significant differences in the lymph flow rates were observed among groups, indicating that CBD treatment had no significant effect on lymph flow in lymph fistula rats.

### 3.2. Lymphatic Isotope Labeled TG and CHOL Output

The TG output was considerably higher in the high-dose CBD group relative to the vehicle group, with a significant increase in the [^3^H]-labeled TG output between the 2nd and 4th hours. At the 4th hour, the hourly lymphatic TG output peaked at ~51% recovery of the infused [^3^H]-TG in the high-dose CBD group, compared to 45% in the low-dose group and 37% in the vehicle group (*p* < 0.05, Figure 2a). Additionally, the TG output AUC was significantly greater in the high-dose CBD group than in the vehicle group (*p* < 0.05, Figure 2c).

In contrast, CHOL output did not significantly differ among the three groups. While the high-dose CBD group showed a trend toward increased [^14^C]-CHOL transport compared to the vehicle group, neither the CHOL output nor its AUC reached statistical significance (Figure 2b,d).

### 3.3. Distribution of Labeled TG and CHOL in the Small Intestine

Several possible explanations exist for the increased [^3^H]-TG in the lymph. One possibility is reduced intestinal retention; however, this is unlikely since CBD treatment did not affect the intestinal distribution of [^3^H]-TG and [^14^C]-CHOL. Specifically, [^3^H]-TG primarily accumulated in the duodenum (M1), with no significant differences among treatment groups (Appendix A). Similarly, [^14^C]-CHOL distribution remained unchanged (Appendix A). Another possibility is increased [^3^H]-TG absorption, but this also seems unlikely, as only very small amounts of [^3^H]-TG and [^14^C]-CHOL were detected in the intestinal lumens, with no significant differences among groups (Appendix A). Minimal reflux into the stomach and negligible colonic recovery further suggest that infused lipids remained within the small intestine during the experiment (Appendix A).

### 3.4. Absorptive and Transport Index

By assessing lipid uptake and secretion efficiency in the three groups, although the adsorptive index for [^3^H]-TG was comparable among groups (Figure 3a), the high-dose CBD group had a significantly higher lymphatic transport index for [^3^H]-TG compared to the vehicle group (Figure 3b). These findings suggest that while TG uptake was unaffected, its incorporation into chylomicrons and subsequent lymphatic transport was enhanced in high-dose CBD-treated rats. Additionally, the lymphatic absorptive and transport indices were similar between the low-dose CBD and vehicle groups. Likewise, none of the three groups showed significant differences in the absorptive or transport indices for [^14^C]-CHOL (Figure 3c,d).

### 3.5. Output of Total TG, CHOL, PL, and NEFA in the Lymph

We also measured the total lymphatic output of TG, CHOL, PL, and NEFA to assess whether CBD treatment alters overall lipid transport and homeostasis. Chemical mass assays showed that lipid levels increased after infusion, peaking at the 5th or 6th hour, with no significant differences among groups (Appendix A). These findings suggest that CBD treatment, at both low and high doses, does not affect lymphatic lipid transport, maintaining normal lipid absorption and homeostasis.

### 3.6. Lymphatic Output of Apolipoproteins

Next, we assessed the lymphatic secretion of ApoA1, ApoA4, and ApoB48 across the three groups. As shown in Figure 4, ApoB48 output was significantly lower in the high-dose CBD group at the 3rd and 6th hours relative to the vehicle group. In contrast, ApoA1 and ApoA4 outputs were significantly higher in the high-dose CBD group between the 4th and 6th hours. Additionally, ApoA4 output in the low-dose CBD group was higher than in the vehicle group at the 6th hour, while ApoA1 output increased between the 4th and 6th hours (*p* < 0.05). No significant difference in the apolipoprotein outputs was found between the high- and low-dose CBD groups.

### 3.7. Expression of Genes Regulating Intestinal Lacteal Structure and Function

To determine whether CBD acutely affected the expression of genes that regulate intestinal lacteal structure and function, we measured the mRNA expression of two key target genes: vascular endothelial growth factor receptor 1 (*Flt1*) and neuropilin1 (*Nrp1*). Both low-and high-dose CBD treatments reduced *Flt1* expression (Figure 5a), while *Nrp1* expression remained unchanged across all groups (Figure 5b).

## 4. Discussion

This study examined whether CBD affects lymphatic lipid output in a conscious rat model with a lymph fistula. After one week of CBD administration, notable effects on lipid transport were observed. Rats receiving a high dose of CBD showed significantly increased [^3^H]-TG output in lymph compared to the vehicle group. Additionally, CBD increased the TG transport index and ApoA1 and ApoA4 lymphatic outputs, while reducing ApoB48 output and *Flt1* gene expression in the small intestine.

The lymph fistula model allows real-time monitoring of intestinal lipid absorption without interference from blood lipases. Its advantages include continuous lymph sampling after lipid infusion, conscious animal conditions free from anesthesia effects, and physiologically relevant samples unaffected by liver metabolism or circulating lipases, making it an optimal model for studying intestinal lipid absorption.

The lymph flow rate remained consistent across all groups, indicating that CBD treatment did not disrupt lymphatic flow or absorptive function. Despite the increased output of [^3^H]-TG and a higher TG transport index, non-labeled TG levels in lymph remained unchanged following CBD treatment, suggesting distinct contributions from endogenous and exogenous lipid sources [16]. The stability of non-labeled TGs, derived from circulating lipoproteins or hepatic sources, further indicates that baseline endogenous lipid metabolism was unaffected. These findings suggest that CBD selectively enhances the transport of newly synthesized or absorbed [^3^H]-TG without altering the baseline pool of endogenous triglycerides. This effect may be driven by CBD-stimulated chylomicron formation or secretion, enriching them with [^3^H]-TG and increasing transport efficiency.

We infused radiolabeled triolein and cholesterol as a single emulsion, yet cholesterol output remained unchanged across the three groups. This discrepancy between triglyceride and cholesterol in lymphatic transport, as well as their differential luminal and mucosal distribution in response to CBD, suggests that the enterocyte uptake, processing, and secretion of these lipids are not always proportionally linked.

Triglyceride and cholesterol follow distinct metabolic pathways within enterocytes before being packaged into chylomicrons (CMs) for secretion into the lymph. Triglycerides are hydrolyzed to monoglycerides and free fatty acids before re-esterification and incorporation into CMs, whereas cholesterol uptake involves specific transporters, such as Niemann-Pick C1-Like 1 (NPC1L1), and requires esterification by acyl-CoA:cholesterol acyltransferase (ACAT). The unchanged cholesterol output suggests that CBD may selectively enhance triglyceride incorporation into CMs without significantly altering cholesterol metabolism, esterification, or efflux pathways.

Lipid absorption predominantly occurs in the small intestine, with the duodenum and jejunum serving as the primary sites of uptake, while the ileum contributes minimally to this process. Consistent with this, our findings showed that the majority of [^3^H]-TG was retained in the proximal small intestinal mucosa across all groups. Furthermore, CBD treatment did not appear to affect gastrointestinal motility, as there were no significant differences in luminal recoveries from the stomach, different segments of the intestine, and the colon across the three experimental groups. CBD also did not significantly alter triglyceride hydrolysis efficiency or its initial uptake in the lumen and mucosa. Notably, minimal regurgitation of the infused emulsion occurred back into the stomach, as indicated by the consistently low levels of recovery for both [^3^H]-TG and [^14^C]-CHOL in the stomach across all groups.

While this study focuses on the effects of CBD on lymphatic lipid transport, it is important to acknowledge potential broader metabolic effects that were not assessed. CBD has been shown to influence glucose metabolism, liver function, and drug metabolism, which could impact lipid processing and overall metabolic balance. Additionally, CBD’s interactions with hepatic enzymes may alter the metabolism of other medications, potentially affecting lipid homeostasis. Future studies should explore these aspects to provide a more comprehensive understanding of CBD’s metabolic effects.

ApoB48 is an essential structural and functional component of CMs, significantly contributing to their assembly and release [17]. As a lipid acceptor, ApoB-48 facilitates the assembly of CMs by stabilizing their lipid core, and it is well established that each CM particle contains a single molecule of ApoB-48 [18]. Given this one-to-one association, lymphatic ApoB48 secretion is commonly used as an indicator for estimating CM particle production by the intestine [19]. Additionally, the TG/ApoB-48 ratio serves as an indicator of CM size, with higher ratios suggesting the formation of larger lipid-rich particles [20].

In this study, we observed that CBD treatment did not alter the secretion of TG-rich CMs in terms of overall TG output. However, high-dose CBD reduced ApoB48 secretion. This suggests that CBD may influence the efficiency of CM assembly or secretion, potentially by modulating ApoB48 availability or lipidation processes. Despite maintained TG transport, the reduction in CM particle number raises the possibility that CBD enhances lipid loading per particle, leading to more lipid-dense CMs.

Interestingly, both low and high doses of CBD increased ApoA4 output in lymph, aligning with the rise in [^3^H]-TG levels. As a lipid-binding protein, apoA4 is primarily secreted in association with nascent CMs into the intestinal lymph following active lipid absorption [19,21]. Lu et al. demonstrated that ApoA4 is essential for enhancing the incorporation of additional lipids into CMs and facilitating the secretion of larger CMs from the intestine [22]. This suggests that increased ApoA4 levels may enhance lipid transport efficiency by optimizing CM lipidation, potentially leading to improved lipid absorption and systemic lipid distribution.

In addition to its involvement in lipid metabolism, ApoA4 has been shown to exert anti-inflammatory and cardioprotective effects. Studies indicate that ApoA4 can inhibit inflammatory cytokine production and reduce oxidative stress [23,24,25], contributing to its protective role in cardiovascular health [26,27]. Furthermore, ApoA4 has been linked to appetite regulation, as it interacts with the central nervous system pathways involved in satiety signaling [28,29]. Increased levels of ApoA4, as observed with CBD treatment in our study, may have broader implications for lipid homeostasis, metabolic health, and inflammation regulation.

Previous studies have demonstrated that ApoA1 is synthesized and secreted by the intestine [30]. Interestingly, both low and high doses of CBD increase ApoA1 output. CMs released by intestinal enterocytes also carry ApoA1, which is rapidly transferred to high-density lipoprotein (HDL) in circulation [31]. Additionally, previous studies have detected HDL in lymph [32], suggesting that intestinally derived ApoA1 may contribute to lymphatic HDL formation.

While this study primarily focuses on triglyceride transport, we acknowledge that other aspects of lipid metabolism, such as systemic lipid profiles, were not extensively analyzed. Notably, we did not observe significant changes in the lymphatic output of cholesterol following CBD treatment. However, the observed increase in ApoA1 and decrease in ApoB48 suggest potential effects on cholesterol metabolism and CM secretion, warranting further investigation. Increased ApoA1 production by CBD could facilitate the efficient assembly and secretion of both CMs and HDL into the lymphatic system, potentially improving lipid transport dynamics. Given that ApoA1 plays a key role in reverse cholesterol transport and possesses anti-inflammatory and antioxidant properties [33,34,35], its upregulation may complement CBD’s therapeutic effects in maintaining intestinal and systemic homeostasis. Future studies should explore whether CBD-mediated ApoA1 induction alters HDL composition and function, as well as its role in modulating inflammation and oxidative stress.

Lipid absorption primarily occurs in the jejunum [36]. Research on the regulation of CM uptake by lacteals is gaining increasing attention. Previous studies have demonstrated that vascular endothelial growth factor A (VEGFA) signaling modulates CM uptake by regulating lacteal cell–cell junctions [10]. As a vital member of the VEGF family, VEGFA is essential for neovascularization and vascular permeability, primarily by interacting with vascular endothelial growth factor receptor 1 (VEGFR-1 or Flt1) [37] and the semaphorin receptor neuropilin-1 (NRP1) [38]. Further research has shown that the absence of these two receptors increases VEGFA bioavailability, enhancing VEGFR2 signaling [34]. This enhanced signaling activates Rac family small GTPase 1 (RAC1), suppressing RhoA activity and reducing Rho-associated kinase (ROCK) activity. Consequently, the suppression of ROCK activity promotes the opening of button-like junctions in lacteals, facilitating chylomicron uptake [39].

In this study, CBD treatment was found to reduce Flt1 expression while having no significant effect on Nrp1 expression. Both Flt1 and Nrp1 play critical roles in vascular endothelial growth factor (VEGF) signaling and vascular regulation; however, their functions in lymphatic transport and lipid metabolism may differ [40]. Flt1 primarily acts as a decoy receptor, modulating VEGF availability and signaling intensity, while Nrp1 serves as a co-receptor that enhances VEGF binding to VEGFR2, influencing endothelial cell function. Reduced Flt1 expression increases VEGFA bioavailability, favoring VEGFR2 signaling, which promotes lymphatic-specific functions such as vessel maintenance and lipid absorption [40]. This receptor shift enhances VEGFR2-driven signaling cascades, which are critical for regulating lacteal permeability and supporting efficient CM transport [39]. The lack of change in Nrp1 expression may indicate that CBD’s effects are more selective toward pathways involving Flt1, rather than broadly affecting VEGF receptor regulation. Further studies are needed to elucidate the mechanistic implications of these differences and their potential impact on lymphatic lipid transport and vascular homeostasis.

While our findings demonstrate that CBD treatment enhances the lymphatic transport of triglycerides and apolipoproteins in adult rats, it is important to acknowledge the species-specific differences in lipid metabolism that may limit direct translation to humans. Unlike humans, who primarily package dietary lipids into chylomicrons, rodents predominantly transport lipids via HDL, which may influence the distribution and metabolism of absorbed lipids. Additionally, differences in lymphatic architecture and triglyceride clearance rates between rodents and humans could impact the extent to which CBD’s effects on lipid absorption and transport are conserved across species. Despite these limitations, rats remain a well-established model for studying intestinal lipid absorption and lymphatic transport, and our findings provide valuable mechanistic insights that warrant further investigation in larger animal models or human clinical studies.

A limitation of this study is the short duration of CBD treatment, as the effects were assessed after only one week of supplementation. It remains uncertain whether these effects persist long term or if compensatory adaptations occur with prolonged exposure. Future studies with extended treatment durations are needed to determine whether CBD’s impact on lymphatic lipid transport and apolipoproteins is sustained over time or subject to metabolic regulation.

## 5. Conclusions

In summary, our study demonstrates that CBD enhances [^3^H]-TG absorption and lymphatic transport in the small intestine. In addition, CBD increases the lymphatic output of ApoA4 and ApoA1, and downregulates the expression of *Flt1* gene in the small intestine. These findings provide new insights into CBD’s contribution to postprandial lipemia and highlight its potential role in modulating lipid metabolism. Moreover, this research offers a foundation for developing CBD-based therapies targeting metabolic and cardiovascular diseases.

Future research should focus on uncovering the mechanisms through which CBD influences lipid absorption. Investigating how CBD modulates the endocannabinoid system and its receptors will be essential in identifying the key signaling pathways involved in lipid metabolism, lipoprotein secretion, and lacteal function. Additionally, exploring the roles of intestinal transporters and lipid metabolic proteins will help determine their contributions to CBD-mediated lipid homeostasis. These efforts will provide a clearer understanding of the molecular basis of CBD’s metabolic effects and may identify potential targets for therapeutic intervention.

## Figures and Tables

**Figure 1 nutrients-17-01034-f001:**
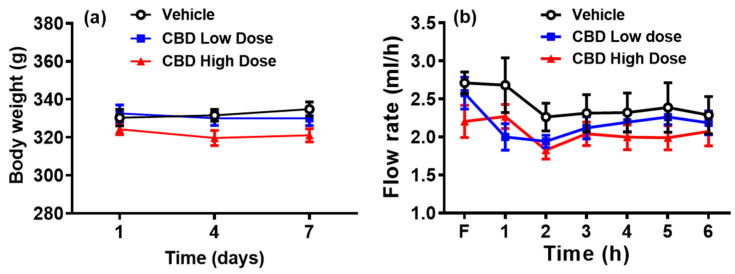
Body weight changes over a 7-day treatment with CBD or vehicle (**a**) and lymph flow rate during the 6 h infusion period (**b**). Mesenteric lymph flow was monitored hourly during the fasting period (F) and for six hours after lipid infusion. Values are means ± SEM. Vehicle group: black circle (n = 8); low-dose CBD group (10 mg/kg): blue square (n = 10); high-dose CBD group (30 mg/kg): red triangle (n = 10).

**Figure 2 nutrients-17-01034-f002:**
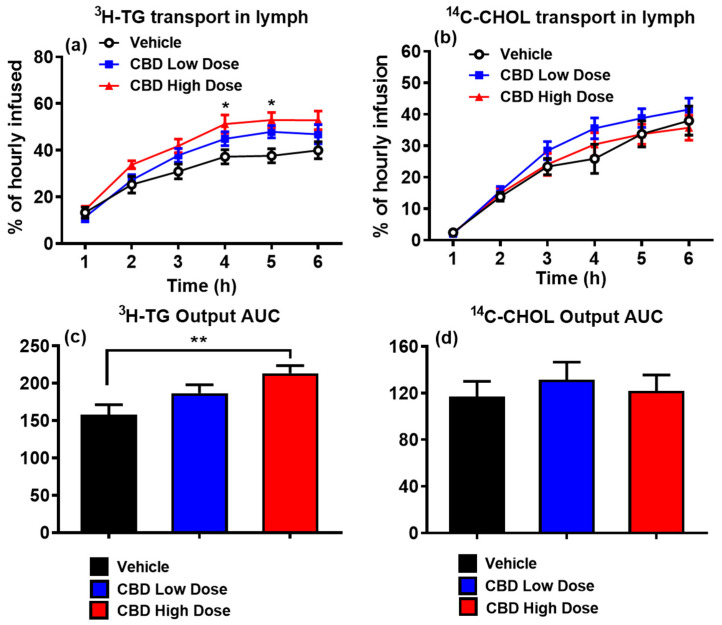
CBD significantly increased TG transport into the lymph of rats. Lymphatic output of [^3^H]-TG (**a**) and [^14^C]-Chol (**b**), expressed as a percentage of the infused hourly dose. Area under the curve (AUC) of TG output (**c**) AUC of CHOL output (**d**). Data are presented as mean ± SEM. Vehicle group: black circles and columns (n = 8); low-dose CBD group (10 mg/kg): blue squares and columns (n = 10); high-dose CBD group (30 mg/kg): red triangles and columns (n = 10). * *p* < 0.05, high-dose CBD vs. vehicle at the same time points for TG levels. ** *p* < 0.01, high-dose CBD vs. vehicle for TG output AUC.

**Figure 3 nutrients-17-01034-f003:**
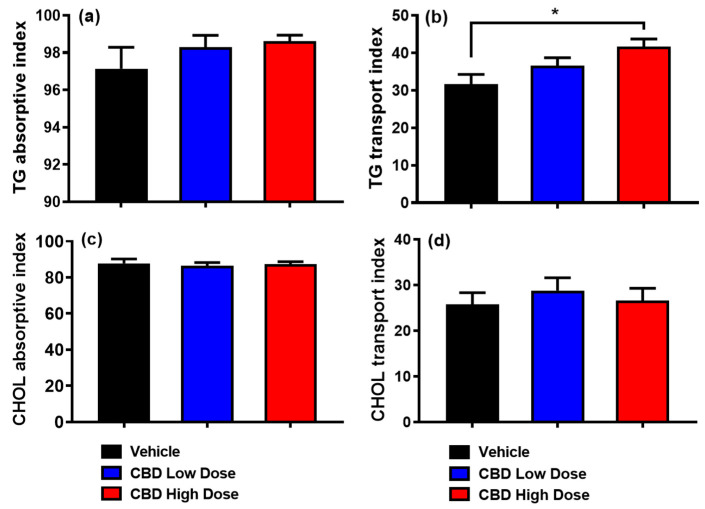
Comparison of absorptive and lymphatic transport indices of [^3^H]-TG and [^14^C]-CHOL across treatment groups. While the absorptive index for [^3^H]-TG was comparable among groups (**a**), the high-dose CBD group had a significantly higher lymphatic transport index for [^3^H]-TG compared to the vehicle group (**b**). No significant differences were observed in the absorptive or transport indices of [^14^C]-CHOL across the three groups (**c**,**d**). Data are presented as mean ± SEM. Vehicle group: black columns (n = 8); low-dose CBD group (10 mg/kg): blue columns (n = 10); high-dose CBD group (30 mg/kg): red columns (n = 10). * *p* < 0.05, high-dose CBD vs. vehicle for TG transport index.

**Figure 4 nutrients-17-01034-f004:**
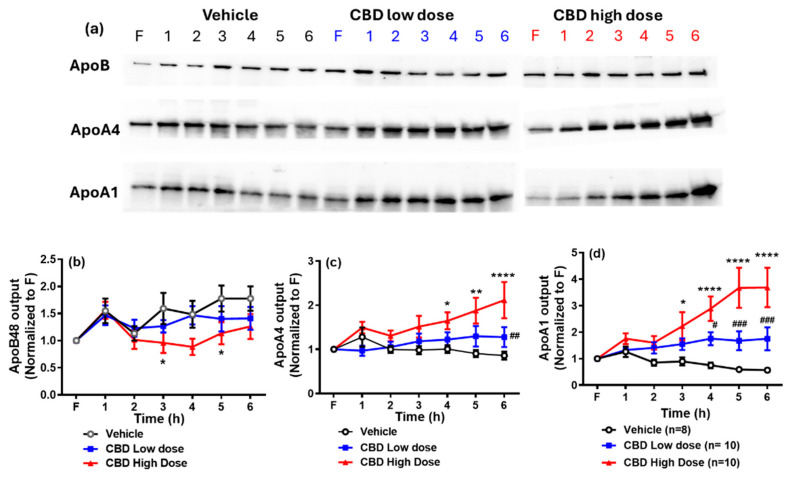
Western blot (**a**) and quantitative analysis of lymphatic ApoB48 (**b**), ApoA4 (**c**), and ApoA1 (**d**) outputs among three groups. The output ratio was calculated through densitometric analysis and quantification of the immunoblots relative to fasting (F) levels. Data are presented as means ± SEM. Vehicle group: black circle (n = 8); low-dose CBD group (10 mg/kg): blue square (n = 10); high-dose CBD group (30 mg/kg): red triangle (n = 10). * *p* < 0.05; ** *p* < 0.01; and **** *p* < 0.0001, comparing apolipoprotein levels between the high-dose CBD and vehicle groups at the same time point. *^#^ p* < 0.05; *^##^ p* < 0.01; *^###^ p* < 0.001, comparing apolipoprotein levels between the low-dose CBD and vehicle groups at the same time points.

**Figure 5 nutrients-17-01034-f005:**
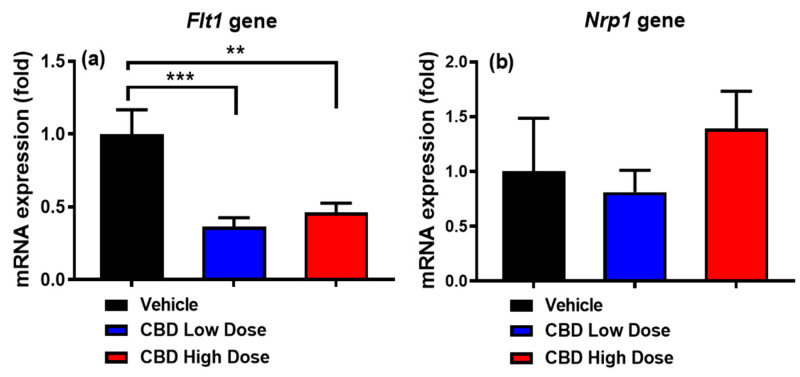
Changes in expression of *Flt1* (**a**) and *Nrp1* (**b**) genes in the jejunum of rats across the three groups. Data are presented as means ± SEM. Vehicle group: black column (n = 8); low-dose CBD group (10 mg/kg): blue column (n = 10); high-dose CBD group (30 mg/kg): red column (n = 10). ** *p* < 0.01, *** *p* < 0.001.

## Data Availability

The data supporting these findings are included in the original manuscript. Further inquiries related to the results can be addressed to the corresponding author.

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
