# Peer review of "The Impact of Cannabidiol (CBD) on Lipid Absorption and Lymphatic Transport in Rats"

_nutrients, 2025, doi:10.3390/nu17061034_

Round 1
Reviewer 1 Report
Comments and Suggestions for Authors
This article examines the effects of cannabidiol (CBD) on lipid absorption and lymphatic transport in rats. The authors use advanced research techniques to understand how CBD may affect lipid metabolism, which is an interesting and relatively new area of research. Their findings suggest that CBD increases the transport of triglycerides into the lymph, influences the secretion of apolipoproteins and regulates the expression of genes related to intestinal lymphatic vascular function. These findings could potentially have implications for the treatment of metabolic diseases such as obesity and dyslipidemia.
Despite its many strengths, the article is not without its limitations.
Strengths of the article
Novel approach to the topic
Cannabidiol (CBD) is a compound that has been extensively studied for its anti-inflammatory, analgesic and neuroprotective effects. However, its effects on lipid metabolism are still relatively poorly understood. The authors address an important and timely topic by investigating how CBD interacts with fat uptake and transport through the lymphatic system. Their findings could contribute to the development of new treatments for metabolic disorders.
Robust methodology
The study was conducted using the lymphatic fistula rat model, which allows direct assessment of fat transport in the lymphatic system. This is an advanced technique that is rarely used in CBD research, which further emphasises the innovative nature of the work.
In addition, the authors used several precise analytical methods, such as Western blot for apolipoprotein analysis, qPCR to assess gene expression and biochemical analysis to measure lipid levels.
This approach allows an accurate assessment of the molecular mechanisms underlying the observed effects.
Clear presentation of the results
The article contains clear diagrams and detailed descriptions of the results so that they are easy to interpret. The authors highlight the main differences between the groups and analyse the effects of CBD on lipid transport, the secretion of apolipoproteins and the changes in the expression of genes that regulate the permeability of intestinal lymphatic vessels.
Although the study was conducted on animals, the results could be important for future research into human lipid metabolism and the possible use of CBD to treat metabolic disorders. This could open the door to new therapies for people suffering from obesity, hyperlipidemia or other diseases associated with impaired fat metabolism.
Weaknesses of the article and areas for improvement
Limitations of the animal model
Although the rat study provides valuable information, the results cannot be directly applied to humans. The lipid metabolism of rodents differs from that of humans, which may lead to different implications in a clinical context. The authors do not sufficiently mention these limitations, which could mislead the reader as to the potential transferability of the results to the human population.
Short study duration
The experiment was conducted over a short period of time (only one week of CBD supplementation). This is an important limitation, as the long-term effects of CBD may differ from those observed in the short term. It is unclear whether the body adapts to the effects of CBD over time or whether these effects persist in the long term.
The article focuses only on the positive effects of CBD, but does not analyse the potential side effects, such as
- The effects of CBD on other aspects of metabolism, such as glucose metabolism
- possible interactions of CBD with other medications,
- effects on liver function, which plays a key role in fat metabolism.
In the absence of this information, the results may appear biassed, which reduces the value of the work as a comprehensive analysis of the effects of CBD on metabolism. Incomplete analysis of lipid transport
The authors focus mainly on triglycerides and neglect other important aspects of lipid metabolism, such as cholesterol transport and its effects on the blood lipid profile. The increase in ApoA1 suggests possible changes in cholesterol metabolism, but a detailed analysis of this phenomenon is lacking. In addition, the effects of CBD have not been compared with other substances that affect lipid transport (e.g. omega-3 acids), which could contribute to a better understanding of CBD's mechanism of action. Unclear interpretation of the apolipoprotein results. The results suggest an increase in ApoA1 and ApoA4 and a decrease in ApoB48 after CBD supplementation, but it is not entirely clear what the long-term consequences of these changes are. An increase in ApoA1 could indicate an improved lipid profile, whereas a decrease in ApoB48 could indicate changes in chylomicron secretion, which requires further analysis.
The main weaknesses are:
- lack of evidence on the long-term effects of CBD,
- incomplete analysis of potential side effects,
- limited reference to cholesterol metabolism.
Reviewer 2 Report
Comments and Suggestions for Authors
Nutrients-3522913
“The Impact of Cannabidiol (CBD) on Lipid Absorption and Lymphatic Transport in Rats”, authored by Qi Zhu, et al..
General comments:
The authors examined the roles of cannabidiol (CBD) on lipid absorption in adult rats. They found that CBD treatment modulates lymphatic lipid composition and apolipoprotein secretion by regulating lymphatic lacteal function, thereby influencing lipid transport and metabolism. The theme of this article is interesting, and it is well-structured. However, some points should be addressed to improve the manuscript.
Specific comments:
- The authors used male adult rats in this experiment. Are there any differences in the results between immature and adult rats, as wells as male and female rats?
- How did the authors determine the dose of CBD? Are the low dose (10 mg/kg) and high dose (30 mg/kg) are reasonable compared to previous reports?
- In the experiment of the lymphatic secretion of apolipoproteins apoA1, apoA4, and apoB48, the authors described that quantification of the immunoblots is normalized by fasting levels. They should show the Western blotting images of the fasting conditions.
- In Fig. 5, it was found that CBD treatments reduced Flt1 expression, while they did not change Nrp1 Although the authors mention about the two factors in the discussion section, they should further discuss about the difference between the expression levels of these two factors.
- The authors consider that CBD treatment modulates lymphatic lipid composition by regulating lymphatic lacteal function. Is it possible to watch the junction transition between button-like and zipper-like formations after the CBD treatment?
Minor:
In line 31, the full spelling of “CB2 receptors” should be provided since it first appeared in this manuscript.

The English could be improved to more clearly express the research.
Reviewer 3 Report
Comments and Suggestions for Authors
This study presents important and novel findings on the effects of CBD on lipid metabolism. The research investigates the impact of CBD on lipid absorption and lymphatic circulation in a rat model, yielding several interesting conclusions. Examining the effects of CBD on lipid metabolism may also be relevant for the treatment of metabolic and cardiovascular diseases. However, it remains unclear how long the effects of CBD persist over time and whether dose-dependent adaptive mechanisms exist. My questions for the authors are as follows:
Can similar effects of CBD be observed in humans?
Is there a threshold beyond which CBD no longer enhances lipid absorption or may even inhibit it?
Does long-term CBD use affect lipid profiles (e.g. HDL/LDL ratio, triglyceride levels)?
How might prolonged CBD treatment influence the gut microbiome?
How does CBD, in combination with other cannabinoids (e.g THC), alter lipid absorption?
What is the impact of long-term CBD use on lipid metabolism and cardiovascular risk factors?
What potential effects could CBD have on lipid metabolism in humans?
The English language of the article is clear and scientifically sound. However, the methodological section could benefit from some rewording to reduce the plagiarism index, which currently exceeds 30%. The references are relevant.
